# Genetic architecture of sleep in a genome wide association study of device measured sleep traits

Laura Portas [1,2] ✉, Hang Yuan [1,2], Lina Cai[3], Karl Smith-Byrne [4], Stefan van Duijvenboden[1,2], Simon D. Kyle[5], David Ray [6,7,8], Joanna MM Howson [3] & Aiden Doherty[1,2]

Sleep is essential for health and regulated by genetic and environmental factors. We perform genome-wide association studies of device-measured sleep duration, efficiency, and accelerometer-derived rapid eye movement (REM) and non-rapid eye movement (NREM) sleep in 80,013 UK Biobank participants. We identify 20 autosomal loci, 12 of which have not been previously reported, including genome-wide significant associations for REM and NREM sleep duration. *MEIS1* shows strong opposing effects on REM and NREM durations and is intolerant to loss-of-function mutations, suggesting an essential role in the regulation of REM/NREM sleep balance. Functional enrichment analysis identifies statistically significant pathways related to chromatin remodelling, lipid metabolism, and metal ion homeostasis while tissue enrichment analysis highlights significant signals in the hypothalamus and frontal cortex. Sex-stratified analyses identify distinct loci, including *FOXP2* and *NRXN3* in females and *LRP1B*, *NPBWR2*, and *PABPC4* in males. Mendelian randomization supports associations between shorter sleep duration and higher cardiometabolic risk. These findings highlight sex- and phase-specific regulators of human sleep architecture, providing biological insights and potential therapeutic targets.

Sleep is essential for overall health and well-being, with disruptions linked to a higher risk of cardiometabolic[1], and mental health conditions[2]. Sleep is regulated by a complex architecture that includes two main phases: rapid eye movement (REM) sleep, which accounts for 20-25% of total sleep, and non-rapid eye movement (NREM) sleep, which makes up 75-80% of total sleep. NREM consists of three stages[3] with distinct brain activity patterns and is associated with restorative processes, while REM is characterised by desynchronised brain activity, muscle atonia, and rapid eye movements, playing a critical role in memory consolidation and emotion regulation. Disruptions to sleep structure, duration, continuity, and timing

can lead to inadequate sleep and are associated with a range of adverse health outcomes[1,2].

Despite the well-described neural circuitry and neurochemistry underpinning sleep state transitions and the coordination between the sleep and circadian systems, the full spectrum of mechanisms governing sleep regulation, particularly the genetic factors influencing sleep traits and their impact on health, remains poorly understood[4]. Sleep behaviours are influenced by a combination of environmental, lifestyle, habitual, and genetic factors, with genetics playing a significant role in the variability of sleep traits. For example, studies have identified associations between genetic variation and sleep duration[5–9],

[1]Big Data Institute, University of Oxford, Oxford, UK. [2]Nuffield Department of Population Health, University of Oxford, Oxford, UK. [3]Human Genetics Centre of Excellence, Novo Nordisk Research Centre, Oxford, UK. [4]Cancer Epidemiology Unit, University of Oxford, Oxford, UK. [5]Sir Jules Thorn Sleep and Circadian Neuroscience Institute, Nuffield Department of Clinical Neurosciences, University of Oxford, Oxford, UK. [6]NIHR Oxford Biomedical Research Centre, John Radcliffe Hospital, Oxford, UK. [7]Oxford Centre for Diabetes, Endocrinology and Metabolism, University of Oxford, Oxford, UK. [8]Oxford Kavli Centre for Nanoscience Discovery, University of Oxford, Oxford, UK. ✉e-mail: laura.portas@ocdem.ox.ac.uk

sleep-wake preferences[10–12], ease of waking[13], and sleep disturbances[14–16], emphasising the heritable nature of sleep homoeostasis.

Previous large-scale genome-wide association studies (GWAS) on sleep traits have been limited by the absence of measurements of REM and NREM sleep. Most earlier GWAS of sleep traits relied on self-reported data, which are prone to bias and lack the precision needed to capture sleep architecture complexity[17]. Recent advances in self-supervised machine learning techniques now make it possible to infer REM and NREM sleep traits from accelerometer datasets, a capacity previously considered unattainable[18]. Leveraging such data provides a valuable opportunity to deepen our understanding of sleep biology, support personalised medicine, and guide targeted interventions for sleep-related disorders.

In this work, we analyse four device-derived sleep traits in a large subset of UK Biobank participants. We identify genetic loci associated with night-time sleep duration, sleep efficiency, and accelerometer-derived REM and NREM sleep, revealing distinct genetic architectures across sleep phases and between sexes.

## Results

### Study sample characteristics

We performed a genome-wide association study (GWAS) of over 9.8 million common variants in 80,013 UK Biobank participants to identify loci associated with four accelerometer-derived sleep traits (Supplementary Fig. 1).

Participants had a mean age of 63 years (standard deviation (SD) = 7.8) and 56% were women (Supplementary Data 1). The cohort was relatively healthy and active, with a mean body mass index (BMI) of 26.6 kg/m² (SD = 4.5), an average daily step count of 9,481, and moderate to high socioeconomic status: 44% reported a high level of education, 57% were never smokers, and 49% consumed alcohol more than three times per week.

On average, participants slept 6.8 h per night (SD = 0.9), including 1.5 h of REM sleep (SD = 0.6) and 5.3 h of NREM sleep (SD = 0.9). These estimates are derived from accelerometer-based inference and, while informative, should be interpreted with caution given the moderate concordance with polysomnography ($\kappa = 0.32$), and the potential for systematic misclassification of sleep stages. Sleep efficiency averaged 82.0%, indicating relatively good sleep continuity. Women tended to have better sleep efficiency but also reported greater use of sleep-influencing medications, particularly antidepressants. Overall, 3.7% of participants reported taking medications that affect sleep (e.g., hypnotics, psychotropics), and 0.1% had documented movement disorders, including restless legs syndrome.

### Heritability, polygenicity, and correlation between sleep traits

SNP-based heritability ($h^2_{SNP}$) estimates for the sleep traits ranged from 9% (sleep efficiency) to 13% (REM and night-time sleep). Genomic inflation was minimal ($\lambda = 1.13$-$1.19$). LD score regression intercepts for genetic correlation analyses were close to 1, with standard errors indicating minimal confounding from sample overlap: 1.0168 (SE = 0.0084) for sleep efficiency with NREM and REM sleep; 1.0145 (SE = 0.0097) for sleep efficiency with REM sleep; and 1.0095 (SE = 0.0069) for all pairings involving night-time sleep. These values suggest that confounding-induced inflation was negligible. The ratio of intercepts indicated that only 4-8% of the observed inflation could be attributed to sources other than polygenicity (Supplementary Fig. 2). Night-time sleep duration showed strong genetic ($r_g = 0.79$, $P < 2.2 \times 10^{-16}$) and phenotypic ($r = 0.79$, $P < 2.2 \times 10^{-16}$) correlation with NREM sleep. Moderate genetic correlations were also observed between night-time sleep and sleep efficiency ($r_g = 0.45$, $P < 2.2 \times 10^{-16}$) and REM sleep ($r_g = 0.43$, $P < 2.2 \times 10^{-16}$), mirrored by phenotypic correlations of $r = 0.42$ ($P < 2.2 \times 10^{-16}$) and $r = 0.36$ ($P < 2.2 \times 10^{-16}$), respectively. Sleep efficiency was genetically correlated with REM ($r_g = 0.22$, $P = 9.5 \times 10^{-6}$)

and NREM sleep ($r_g = 0.35$, $P = 1.9 \times 10^{-12}$), with corresponding phenotypic correlations of $r = 0.18$ ($P < 2.2 \times 10^{-16}$) and $r = 0.31$ ($P < 2.2 \times 10^{-16}$), respectively. Finally, REM and NREM sleep showed modest negative genetic ($r_g = -0.21$, $P = 1.7 \times 10^{-5}$) and phenotypic ($r = -0.29$; $P < 2.2 \times 10^{-16}$) correlations (Supplementary Fig. 3). Scatter plots of night-time sleep, REM, and NREM durations confirmed the observed phenotypic correlations. The negative correlation between REM and NREM sleep was not driven by individuals with extreme sleep durations, but was consistent across the distribution (Supplementary Fig. 4).

### SNP-level analysis

We identified 20 independent autosomal loci associated with sleep traits at genome-wide significance ($P < 5.0 \times 10^{-8}$), including 12 signals not previously reported for these sleep traits (Table 1; Fig. 1; Supplementary Figs. 5 and 6; Supplementary Data 2). Six loci were associated with night-time sleep duration, three of which have not been previously reported for this trait (2p16.1, 6p22.3, 7q31.1). For sleep efficiency, one locus was identified at 7q11.22, previously linked to total sleep duration but not sleep efficiency. Here, we performed genome-wide association analyses of REM and NREM sleep and identified five loci for REM sleep (1p21.3, 3p11.1, 11q13.2, 11q13.4, and 22q13.1) and three for NREM sleep (13q14.2, 14q22.3, and 15q23). According to our predefined criteria, these loci have not been previously reported for REM or NREM sleep. We note, however, that the 15q23 locus maps to MAP2K5, a gene previously implicated in restless legs syndrome (RLS), suggesting shared biology between RLS and NREM sleep regulation. A signal at 8q24.3 was associated with both REM and NREM sleep. No significant associations were detected on chromosome X. Multi-trait analysis of GWAS (MTAG) did not yield additional loci beyond those identified in single-trait analyses, likely due to modest genetic correlations between traits and the high power of the individual GWAS (Supplementary Data 3; Supplementary Fig. 3).

Conditional analyses identified additional independent signals within several loci. At 2p14, two intronic variants in MEIS1, rs4544423 ($\beta = 1.13$ min/night; $P = 1.9 \times 10^{-10}$) and rs182588061 ($\beta = 3.81$ min/night; $P = 1.2 \times 10^{-8}$), remained significantly associated with REM sleep after adjusting for the lead SNP rs113851554 ($\beta = 9.05$ min/night; $P = 1.2 \times 10^{-116}$). Similarly, at 8q24.3, a secondary intronic variant in KCNK9 (rs888346; $\beta = 0.98$ min/night; $P = 3.3 \times 10^{-8}$) was identified after conditioning on rs2542425 ($\beta = 1.21$ min/night; $P = 6.7 \times 10^{-12}$). In addition, fine-mapping across all loci highlighted likely causal variants with high posterior probability of causality (PIP ≥ 95%) (Table 1 and Supplementary Data 4). At 2p14, the intronic variant rs113851554 in MEIS1 was associated with both REM ($\beta = 9.05$ min/night; $P = 1.2 \times 10^{-116}$) and NREM sleep ($\beta = -6.92$ min/night; $P = 1.6 \times 10^{-30}$) and had a posterior probability of causality (PIP) of 100%.

At 11q13.4, the intronic variant rs7116582 in C2CD3 (PIP = 95%), was associated with REM sleep ($\beta = -3.07$ min/night; $P = 1.7 \times 10^{-8}$).

Sensitivity analyses excluding individuals with documented movement disorders or those using medications known to affect sleep, as well as models adjusted for BMI, did not materially change the top associations. We further excluded individuals with night-time sleep duration <7 h of sleep, thereby reducing the potential influence of severe short-sleep cases, who may be enriched for undiagnosed RLS; associations at MEIS1 and other loci remained directionally consistent, although some signals, especially for night-time sleep duration, lost genome-wide significance due to reduced statistical power (Supplementary Data 4 and Supplementary Figs. 7–8). In addition, to address the underestimation of RLS in the UK Biobank, we excluded individuals with ICD-10-coded RLS/movement disorders, as well as probable RLS cases identified through the online sleep questionnaire (collected 2022–2024). This removed 6561 individuals (≈ 8.2% of the cohort), a prevalence consistent with epidemiological estimates of RLS in European populations. Associations at MEIS1 and other key loci persisted

**Table 1 | Genome-wide significant (_P_ < 5 × 10⁻⁸) loci associated with sleep traits in subjects of European ancestry in the UK Biobank**

| Locus | Lead SNP | Position (GRCh37) NCBI 37 | Most severe consequence | Nearest Gene(s)§ | E/A(E/A) | EAF | Info | Beta | SE | _P_ | Most likely causal SNP (PIP) |
|---|---|---|---|---|---|---|---|---|---|---|---|
| Night-time sleep duration, minutes/day | | | | | | | | | | | |
| 2p16.1 | rs11125769 | 59,310,698 | intergenic variant | _FANCL_ | A/C | 0.645 | 1.00 | −1.705 | 0.294 | 6.51E-09 | rs11125769 (0.24) |
| 2q14.1 | rs199993536 | 114,082,628 | intergenic variant | _PAX8_ | T/TA | 0.782 | 0.98 | −3.297 | 0.341 | 4.14E-22 | rs199993536 (0.43) |
| 6p22.3 | rs112507172 | 19,096,689 | intergenic variant | _RNF144B_ | A/G | 0.752 | 1.00 | 1.954 | 0.324 | 1.73E-09 | rs112507172 (0.22) |
| 6p22.2 | rs6940638 | 27,046,250 | upstream gene variant | _H2BC11_ | A/G | 0.771 | 1.00 | 1.813 | 0.331 | 4.54E-08 | rs79887176 (0.08) |
| 7q31.1 | rs13237149 | 113,871,993 | intron variant | _FOXP2_ | T/A | 0.686 | 0.98 | −1.794 | 0.305 | 4.27E-09 | rs13237149 (0.62) |
| 16q12.2 | rs62033413 | 53,830,055 | intron variant | _FTO_ | C/G | 0.585 | 1.00 | 1.568 | 0.284 | 3.39E-08 | rs186842218 (0.06) |
| Sleep efficiency, % | | | | | | | | | | | |
| 2p16.1 | rs11125769 | 59,310,698 | intergenic variant | _FANCL_ | A/C | 0.645 | 1.00 | −1.705 | 0.294 | 6.51E-09 | rs11125769 (0.24) |
| 7q11.22 | rs2006810 | 69,902,152 | intron variant | AUTS2 | T/C | 0.604 | 0.99 | 0.245 | 0.044 | 1.76E-08 | rs3113257 (0.14) |
| REM sleep, minutes/day | | | | | | | | | | | |
| 1p21.3 | rs11165579 | 96,759,720 | intron variant | _LINC01787_ | A/G | 0.706 | 1.00 | 1.124 | 0.193 | 5.42E-09 | rs11165579 (0.09) |
| 1p21.3 | rs11165579 | 96,759,720 | intron variant | _LINC01787_ | A/G | 0.706 | 1.00 | 1.124 | 0.193 | 5.42E-09 | rs11165579 (0.09) |
| 2p14 | rs113851554 | 66,750,564 | intron variant | _MEIS1_ | G/T | 0.943 | 0.93 | 9.051 | 0.394 | 1.22E-116 | rs113851554 (1.00) |
| | rs4544423* | 66,750,017 | intron variant | _MEIS1_ | T/G | 0.404 | 1.00 | 1.126 | 0.177 | 1.94E-10 | |
| | rs182588061* | 66,757,709 | intron variant | _MEIS1_ | G/T | 0.990 | 0.83 | 3.812 | 0.670 | 1.25E-08 | |
| 2q14.1 | rs199993536 | 114,082,628 | intergenic variant | _PAX8_ | T/TA | 0.782 | 0.98 | −1.784 | 0.215 | 9.86E-17 | rs199993536 (0.27) |
| 3p11.1 | rs6779623 | 87,849,642 | intergenic variant | _HTR1F_ | G/C | 0.705 | 0.99 | 1.178 | 0.193 | 1.11E-09 | rs6779623 (0.20) |
| 6p21.2 | rs4714163 | 38,438,771 | intron variant | _BTBD9_ | T/C | 0.708 | 1.00 | −1.566 | 0.193 | 5.48E-16 | rs4714163 (0.38) |
| 8q24.3 | rs2542425 | 140,632,384 | intron variant | _KCNK9_ | T/C | 0.554 | 1.00 | 1.214 | 0.177 | 6.66E-12 | rs888346 (0.64) |
| | rs888346* | 140,713,897 | intron variant | _KCNK9_ | C/T | 0.474 | 0.98 | 0.980 | 0.177 | 3.32E-08 | |
| 11q13.2 | rs10680443 | 66,685,778 | intron variant | _PC_ | G/GGA | 0.646 | 0.99 | −1.021 | 0.185 | 3.24E-08 | rs10680443 (0.16) |
| 11q13.4 | rs7116582 | 73,732,100 | intron variant | _C2CD3_ | G/A | 0.974 | 1.00 | −3.075 | 0.545 | 1.72E-08 | rs7116582 (0.95) |
| 16q12.2 | rs62033400 | 53,811,788 | intron variant | _FTO_ | A/G | 0.603 | 1.00 | 1.003 | 0.180 | 2.33E-08 | rs62033417 (0.08) |
| 22q13.1 | rs6001802 | 40,543,215 | intron variant | _TNRC6B_ | G/A | 0.745 | 0.98 | −1.116 | 0.203 | 3.71E-08 | rs761126143 (0.08) |
| Non-REM sleep, minutes/day | | | | | | | | | | | |
| 2p14 | rs113851554 | 66,750,564 | intron variant | _MEIS1_ | G/T | 0.943 | 0.93 | −6.917 | 0.602 | 1.58E-30 | rs113851554 (1.00) |
| 2p14 | rs113851554 | 66,750,564 | intron variant | _MEIS1_ | G/T | 0.943 | 0.93 | −6.917 | 0.602 | 1.58E-30 | rs113851554 (1.00) |
| 6p22.2 | rs12215241 | 27,023,081 | intergenic variant | _H2BC11_ | G/A | 0.772 | 1.00 | 2.052 | 0.319 | 1.28E-10 | rs371182726 (0.19) |
| 8q24.3 | rs2542423 | 140,635,688 | intron variant | _KCNK9_ | G/C | 0.556 | 0.99 | −1.571 | 0.272 | 7.49E-09 | rs2542423 (0.47) |
| 13q14.2 | rs7491366 | 47,928,776 | intergenic variant | _HTR2A_ | T/C | 0.661 | 0.99 | 1.610 | 0.285 | 1.65E-08 | rs7491366 (0.21) |
| 14q22.3 | rs1209087 | 55,493,220 | intron variant | _WDHD1_ | C/T | 0.429 | 0.99 | −1.502 | 0.273 | 3.67E-08 | rs1882747021 (0.05) |
| 14q32.2 | rs7146019 | 99,734,984 | intron variant | _BCL11B_ | A/G | 0.429 | 0.97 | −1.708 | 0.277 | 7.12E-10 | rs7146019 (0.23) |
| 15q23 | rs12902804 | 67,988,658 | intron variant | _MAP2K5_ | G/A | 0.503 | 1.00 | −1.504 | 0.270 | 2.42E-08 | rs12902804 (0.08) |
| 19p13.2 | rs12973413 | 9,942,222 | upstream gene variant | _UBL5_ | A/G | 0.447 | 0.98 | −1.554 | 0.272 | 1.13E-08 | rs1292220483 (0.40) |

P values are two-sided.

_E_ effect allele, _A_ alternative allele, _Chr_ chromosome, _EAF_ effect allele frequency, _INFO_ imputation quality, _PIP_ posterior inclusion probability. Analyses were adjusted for age, sex, centre, season, genetic ancestry and genotyping array. §Distance to transcription start site (TSS) for non-intronic variants. *Secondary signals from conditional analysis.

with very similar effect sizes (Supplementary Data 5), supporting that the observed signals are not driven by diagnosed or probable RLS.

**Sex-specific genetic effects**

Sex-stratified GWAS identified loci with sex-specific associations, suggesting sex-based regulation of sleep traits (Supplementary Data 4; Supplementary Fig. 9). Genome-wide SNP heritability estimates were broadly similar between males and females across all traits: sleep efficiency ($h^2$ = 0.080 [SE = 0.015] in males, 0.099 [0.012] in females), NREM sleep (0.141 [0.015] vs 0.130 [0.016]), REM sleep (0.126 [0.015] vs 0.121 [0.015]), and night-time sleep duration (0.123 [0.014] vs 0.130

[0.013]). Cross-sex genetic correlations were high for sleep efficiency ($r_g$ = 0.98), REM sleep (1.00), and night-time sleep duration (0.97), consistent with largely shared polygenic architecture. In contrast, NREM sleep showed a lower correlation ($r_g$ = 0.84), suggesting partial sex-specific differences in its genetic determinants.

Four loci showed significant sex heterogeneity: 7q31.1 for night-time sleep ($I^2$ = 91.20, $P$ = 0.001), and 1p21.3 ($I^2$ = 79.80, $P$ = 0.026), 2p14 ($I^2$ = 96.60, $P$ = 6.47 × 10⁻⁸), and 6p21.2 ($I^2$ = 83.20, $P$ = 0.015) for REM sleep. Additional sexually dimorphic loci, not genome-wide significance in the overall analysis, were also identified (Supplementary Data 6). In females, an intergenic variant (14:78496241_CA_C) was

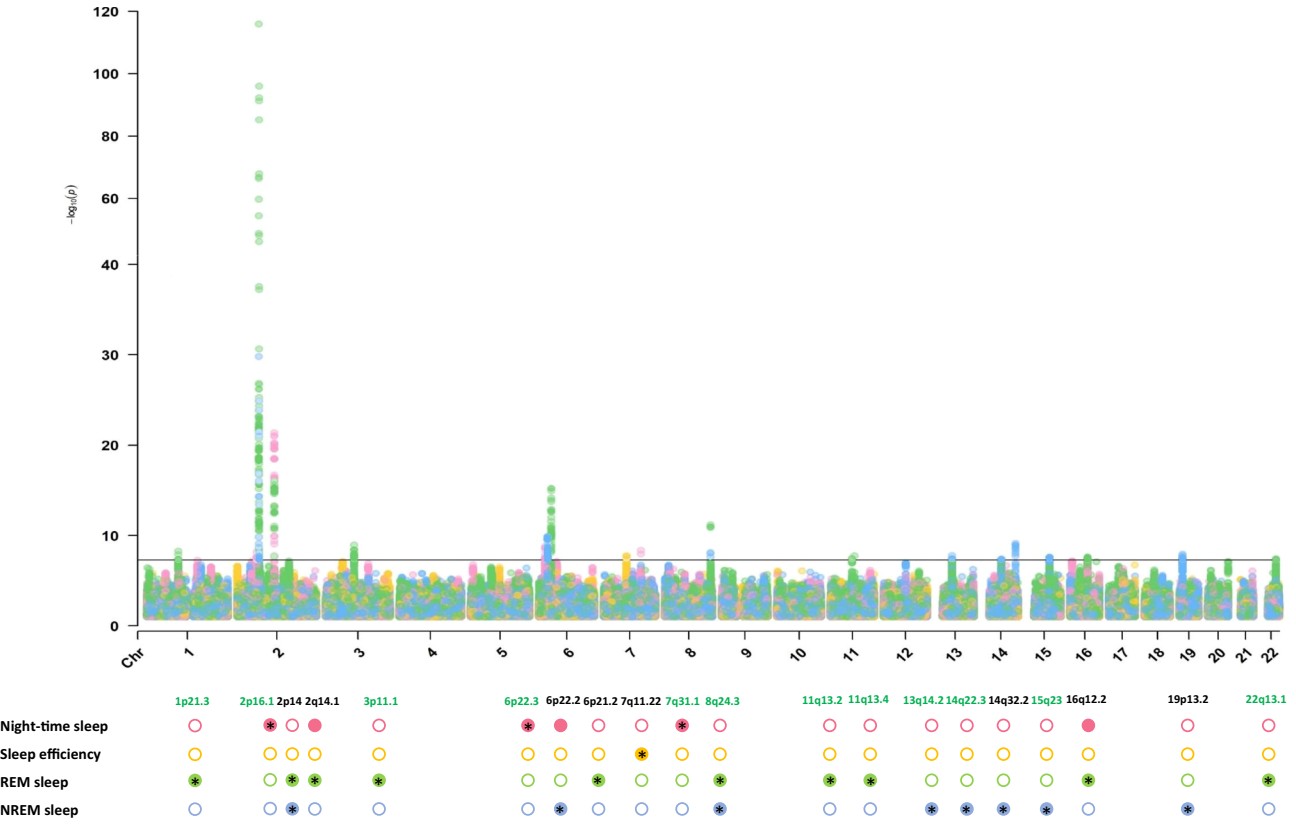

**Fig. 1 | Combined Manhattan plot of GWAS results for the four device-measured sleep traits in 80,013 UK Biobank participants.** Genome-wide association results are shown for night-time sleep duration, sleep efficiency, REM and NREM sleep. Genome-wide significance was defined as $P < 5 \times 10^{-8}$.

significantly associated with night-time sleep duration ($\beta = 3.63$ min/night; $P = 4.16 \times 10^{-8}$) and lies -140 kb from the *NRXN3*. Another female-specific intergenic variant (4:104463040_AT_A), was associated with sleep efficiency ($\beta = 38.6\%$; $P = 4.0^{-8}$), and is located -178 kb from *TACR3*. In males, three intronic variants showed significant associations with REM sleep: rs568010150 in *LRP1B* ($\beta = -1.96$ min/night; $P = 1.1 \times 10^{-8}$), rs398111 in *NPBWR2* ($\beta = 1.71$ min/night; $P = 3.3 \times 10^{-8}$), and rs1180331 in *PABPC4* ($\beta = -1.58$ min/night; $P = 1.1 \times 10^{-8}$). For NREM sleep, a female-specific association was found for rs1799252 in *FOXP2* ($\beta = -2.16$ min/night; $P = 6.4 \times 10^{-9}$), while an intergenic variant (rs6465150) was associated in males ($\beta = -3.69$ min/night; $P = 2.4 \times 10^{-8}$), and is located -78 kb from the *ZNF804B* gene. Given the mean age of female participants (62.1 years), most were likely post-menopausal, and 38.2% reported ever using hormone replacement therapy (HRT), which may partly contribute to the observed sex-specific associations. These findings highlight sex-specific genetic contributions to sleep traits and suggest distinct biological pathways. However, given the reduced sample sizes, these sex-stratified results were not subjected to the fine-mapping and downstream functional prioritisation pipeline used in the main analyses and should be considered exploratory, warranting replication in larger cohorts.

## Gene-level analysis
Gene-based analyses revealed significant enrichment in pathways related to chromatin organisation, metal ion homoeostasis (particularly copper, zinc, cadmium), and lipid metabolism (Supplementary Data 7). Given that iron deficiency is a well-established risk factor for restless legs syndrome and magnesium supplementation is frequently used by patients as an informal sleep aid, the enrichment of metal ion homeostasis is of particular biological relevance (see Discussion).

Chromatin-related pathways such as nucleosome assembly and DNA-protein interactions were enriched across all sleep traits, while fatty acid metabolism showed a specific association with REM sleep.

Tissue enrichment analysis highlighted multiple brain regions for night-time sleep (e.g., cerebellum, cortex, basal ganglia, hypothalamus, amygdala) and the pituitary gland, suggesting a role for neuroendocrine regulation (Supplementary Fig. 10). REM sleep was enriched in the frontal cortex, general cortex, and cingulate cortex, while NREM sleep showed enrichment in the cerebellum and nucleus accumbens.

No cell types reached significance after Bonferroni correction for multiple testing (Supplementary Data 8). Nominal enrichments were observed for GABAergic neurons in NREM sleep and excitatory cortical neurons in REM sleep, consistent with the neurobiology of sleep stage regulation, although these findings should be interpreted with caution.

Gene-level aggregation also identified ten additional trait-associated loci beyond SNP-level analysis. These included one locus associated with night-time sleep duration (near *SGCZ*), three with sleep efficiency (near *FAM150B*, *PBRM1*, and *ITGA2B*), four with REM sleep (near *CAMTA1*, *PKP4*, *SKOR1*, and *TBX6*), and two with NREM sleep (near *SGCZ* and *CAND1*) (Supplementary Data 9).

## Chromatin interaction, eQTL mapping, and eQTL colocalization analysis
Integrated analyses identified loci associated with neurological, metabolic, gastrointestinal, and immune-related genes (Supplementary Data 10 and 11, Supplementary Figs. 11–12).

Chromatin interaction mapping linked several loci to candidate genes, including *PSD4*, *PAX8*, *FTO*, and *IRX3* for night-time sleep, and *MEIS1*, *HTR1F*, and *HTR2A* for REM and NREM sleep.

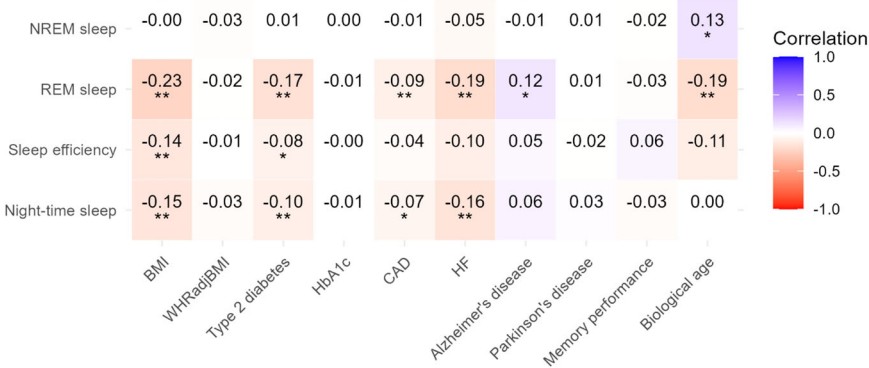

**Fig. 2 | Genetic correlations of sleep traits with other phenotypes.** The genetic correlation coefficient was calculated using LD score regression (LDSC) and is denoted by a colour scale from −1 (red; negatively correlated) to +1 (blue; positively correlated). A single asterisk indicates two-sided $P \leq 0.05$ and a double asterisk indicates two-sided $P \leq 0.01$. BMI body mass index, WHRadjBMI, waist-hip ratio adjusted for BMI, HbA1c haemoglobin A1c, CAD coronary artery disease, HF heart failure.

eQTL mapping revealed tissue-specific gene expression patterns, and colocalization analysis supported regulatory roles for several key genes. For night-time sleep, colocalization at 2q14.1 with *FOXD4L1* and *CBWD2* in thyroid tissue suggests a neuroendocrine component, in line with prior associations at PAX8, a thyroid lineage transcription factor identified in early sleep GWAS. For REM sleep, *HTR1F* colocalized in spleen, thyroid, adipose, and heart tissue, reflecting potential immune-metabolic contributions. For NREM sleep, relevant colocalizations included *BTN2A2* (within the MHC region), *WDHD1*, and *PIN1-DT*, with signals in pancreas, oesophageal mucosa, and frontal cortex. Although the MHC locus requires cautious interpretation due to its complex LD structure, *BTN2A2*, while related to the immunoglobulin genes, has distinct roles in lipid metabolism, further emphasising links between immune and metabolic functions in deep, restorative sleep. To further validate the biological interpretation of these loci in the central nervous system, we interrogated independent brain-specific eQTL resources (PsychENCODE, CMC, BrainSeq, BRAINEAC, and xQTLServer). Several key signals replicated across these datasets, including robust cis-eQTL effects for MEIS1 (REM and NREM sleep), and associations at the extended BTN3A/BTN2A1 locus (night-time and NREM sleep) in DLPFC and bulk cortex. Additional evidence was observed for HTR1F, PC, ZNF391, and SKOR1 across multiple brain panels (Supplementary Data 12). These concordant results strengthen the evidence that regulatory variation in brain-expressed genes contributes to sleep phenotypes, beyond the GTEx-derived associations.

No significant sex-specific colocalizations were detected. Variants such as rs6779623 and rs56169023 regulate *HTR1F* expression in peripheral tissues, suggesting possible non-neuronal contributions to sleep physiology. However, publicly available eQTL data (e.g., FiveX, BrainSeq) also show regulatory activity in brain tissues, and suggest that these variants may influence additional genes beyond *HTR1F*. This broader regulatory context raises the possibility of both central and peripheral contributions to REM sleep regulation. Similarly, genes such as *WDHD1* and *SKOR1*, enriched in gastrointestinal and metabolic tissues, reinforce the idea that peripheral systems contribute substantially to NREM sleep.

## Sleep traits and health outcomes

LD score regression (LDSC) revealed significant genetic correlation between sleep traits and various health outcomes (Fig. 2). Night-time sleep was negatively correlated with BMI, type 2 diabetes mellitus (T2DM), coronary artery disease (CAD), and heart failure (HF), while sleep efficiency was negatively correlated with BMI and T2DM. REM sleep showed negative genetic correlations with estimated genetic risk of several cardiometabolic traits (HF, CAD, T2DM, BMI) and with

biological age, but a positive correlation with genetic risk of Alzheimer's disease. In contrast, NREM sleep was positively associated with biological age.

Pairwise colocalization analyses identified shared loci between sleep traits and T2DM, notably at 16q12.2 (FTO, rs55872725, PP.H4 = 0.83 for night-time sleep and PP.H4 = 0.82 for REM sleep), and 22q13.1 (TNRC6B, rs6001802, PP.H4 = 0.80 for REM sleep), including genes previously linked to chronotype. Given previous associations of these loci with obesity, we performed additional analyses using BMI-adjusted summary statistics to assess potential confounding. Adjustment for BMI did not materially affect the association at 22q13.1 (rs6001802, PP.H4 = 0.79 for REM sleep), but at 16q12.2, the colocalization signal weakened (decrease in PP.H4, increase in PP.H3), suggesting partial mediation by BMI. Similar patterns were observed for other traits: colocalization signals at 16q12.2 for BMI (rs1558902, PP.H4 = 0.85 for night-time sleep and PP.H4 = 0.84 for REM sleep) and heart failure (16:53803223_G_A, PP.H4 = 0.85 for night-time sleep and PP.H4 = 0.87 for REM sleep) was reduced after adjustment (PP.H4 < 0.5). Likewise, the association with HbA1c (16:53798523_G_A, PP.H4 = 0.75 for night-time sleep) was no longer supported post-adjustment.

We performed 40 bidirectional MR analyses (4 sleep traits × 10 outcomes), and applied Bonferroni correction to account for multiple comparisons ($P < 0.012$ for sleep → outcome; $P < 0.005$ for outcome → sleep). All P-values are reported unadjusted; interpretations reflect these corrected thresholds. Mendelian randomisation analyses (Supplementary Data 13; Fig. 3; Supplementary Fig. 13) supported robust associations of longer night-time sleep with lower BMI ($\beta = -0.366$ per hour, se = 0.084, $P = 7.4 \times 10^{-6}$), HbA1c ($\beta = -0.090$, se = 0.024, $P = 5.7 \times 10^{-5}$), and biological age ($\beta = -0.414$, se = 0.120, $P = 7.6 \times 10^{-4}$), all of which remained significant after Bonferroni correction ($P < 0.012$). The association with T2DM risk (Odds Ratio (OR) = 0.610, 95% Confidence Interval (CI) = 0.415–0.894, $P = 0.011$), was statistically significant, albeit close to the Bonferroni correction threshold. Longer REM sleep showed suggestive associations ($P < 0.05$) with improved memory ($\beta = 0.036$, se = 0.018, $P = 0.014$), lower HF (OR = 0.710, 95% CI = 0.529–0.953, $P = 0.020$), CAD risk (OR = 0.845, 95% CI = 0.717–0.997, $P = 0.043$), and reduced HbA1c ($\beta = -0.084$, se = 0.036, $P = 0.015$), although none of these passed the Bonferroni-corrected threshold ($P < 0.012$). The association with higher waist-to-hip ratio adjusted for BMI (WHRadjBMI) ($\beta = 0.072$, se = 0.024, $P = 0.003$) remained statistically significant after correction. Higher sleep efficiency was associated with lower BMI ($\beta = -2.322$, se = 0.692, $P = 8.00 \times 10^{-4}$ per 1% increase) and higher WHRadjBMI ($\beta = 2.322$, se = 0.733, $P = 0.001$), both of which remained statistically significant after

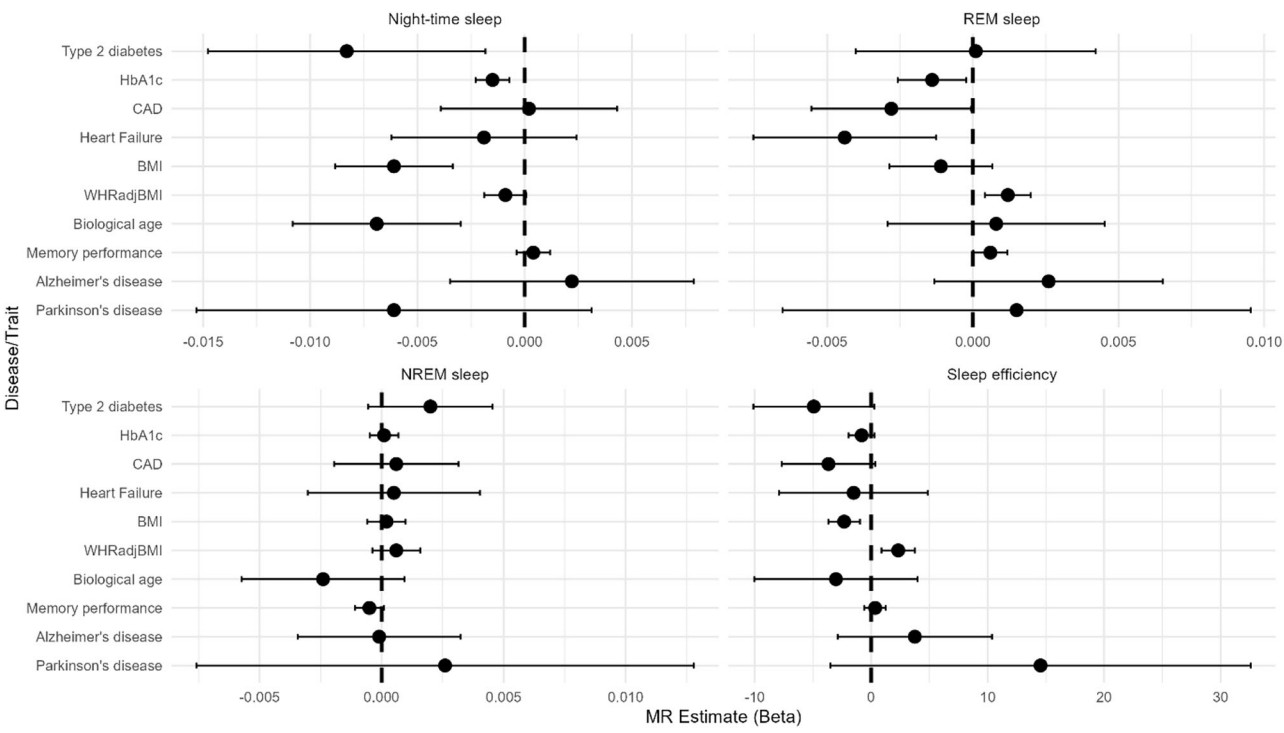

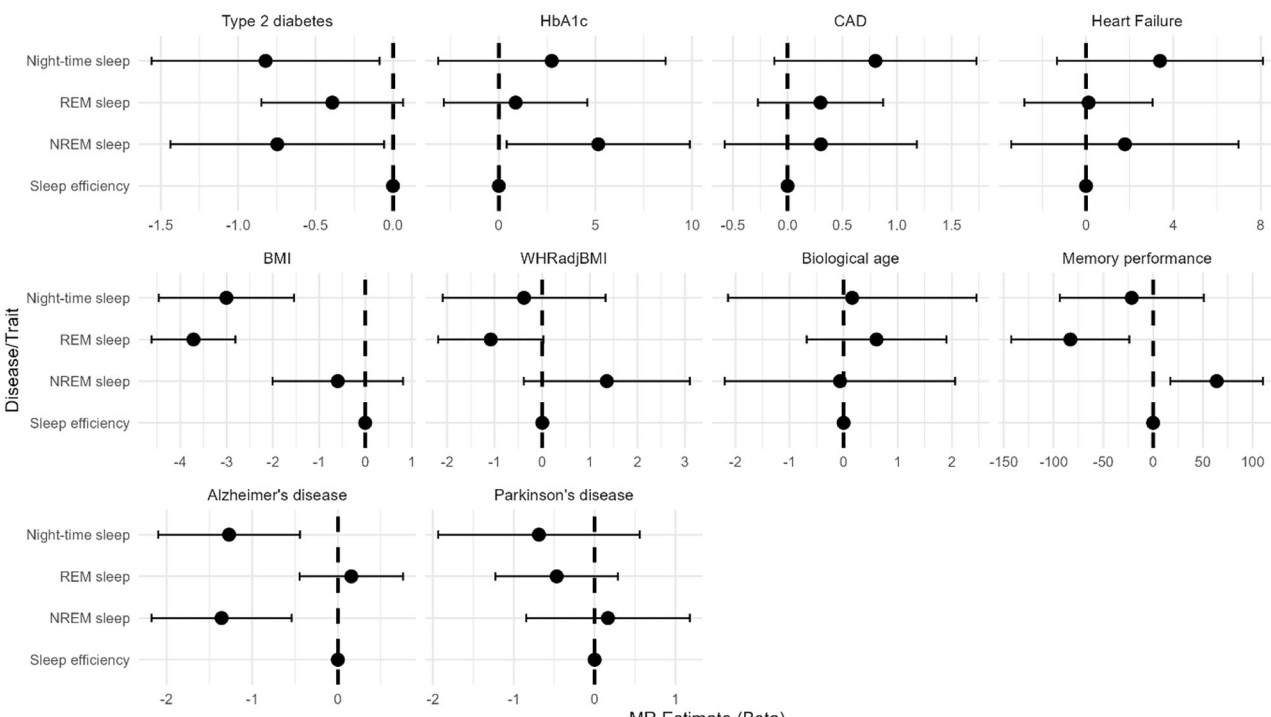

**Fig. 3 | Bidirectional Mendelian randomisation results: causal effects of sleep traits on health outcomes and vice versa.** Results correspond to the inverse-variance weighted (IVW) method with random effects. Traits shown at the top of each panel represent the exposure in the Mendelian randomisation analysis. Error bars indicate 95% confidence intervals. The number of genetic instruments used in each MR analysis is indicated in Supplementary Data 13. P-values are two-sided; statistical significance is indicated as described in the Methods. Analyses are based on independent human participants (biological replicates), with the individual participant as the unit of study. HbA1c haemoglobin A1c, CAD coronary artery disease, BMI body mass index, WHRadjBMI waist-hip ratio adjusted for BMI.

Bonferroni correction. Reverse MR indicated that higher BMI causally reduced REM ($\beta = -3.716$ min/night per kg/m$^2$, se = 0.463, $P = 1.0 \times 10^{-15}$), night-time sleep ($\beta = -3.004$ min/night per kg/m$^2$, se = 0.746, $P = 5.6 \times 10^{-5}$), and sleep efficiency ($\beta = -0.490\%$ per kg/m$^2$, se = 0.110, $P = 5.0 \times 10^{-6}$); all associations remained significant after Bonferroni

correction ($P < 0.005$ for outcomes as exposures). Higher Alzheimer's risk was associated with shorter night-time sleep ($\beta = -1.271$ min/night, se = 0.423, $P = 0.003$) and NREM sleep ($\beta = -1.358$ min/night, se = 0.418, $P = 0.001$), both of which passed the Bonferroni correction threshold ($P < 0.005$ for reverse MR).

We found suggestive evidence ($P < 0.05$, not significant after correction) that higher risk of type 2 diabetes may be associated with lower NREM sleep duration (β = −0.747 min/night, se = 0.352, $P = 0.034$), and that better memory performance may be linked to shorter REM sleep (β = −83.11 min/night, se = 30.28, $P = 0.006$) and longer NREM sleep (β = 63.98 min/night, se = 23.75, $P = 0.007$).

## Discussion

In this genome-wide association study of over 80,000 UK Biobank participants, we investigated four accelerometer-derived sleep traits, including REM and NREM sleep traits. We identified 20 autosomal loci, including 12 loci not previously reported for REM and NREM sleep. These findings reveal sleep-phase-specific genetic contributions to human sleep architecture and suggest links between sleep traits and cardiometabolic outcomes.

Compared with the earlier GWAS of accelerometer-derived sleep traits in UK Biobank (Jones et al. 2019), our study shows both overlap and divergence. Several loci were replicated across studies, but we observed lower heritability estimates and differences in mean trait values (-0.5 h shorter sleep duration and -4% lower efficiency). These discrepancies primarily reflect differences in phenotype definition: we used a deep-learning model trained on polysomnography to derive REM and NREM sleep, and applied stricter exclusions (e.g., shift workers), thereby improving biological specificity at the cost of lower global heritability. Notably, MEIS1 − previously a top hit for sleep duration and efficiency − did not replicate for these traits, but instead emerged with strong, opposing effects on REM and NREM sleep in our analysis, highlighting the added resolution of phase-specific phenotyping. Furthermore, we implemented multiple sensitivity analyses that were not undertaken in the earlier work, reinforcing the robustness of our findings. Taken together, these contrasts indicate that differences in study design and phenotype definition contribute to both overlaps and previously unreported findings, particularly in resolving stage-specific genetic influences on sleep. We identified an association at the 3p11.1 locus with REM sleep, with chromatin and eQTL analyses highlighting *HTR1F* as a regulatory candidate. This gene encodes a serotonin receptor and shows eQTL signals in metabolically relevant tissues, including adipose, heart, and pancreas. While serotonin's role in sleep regulation is well established, our findings suggest that *HTR1F* may also contribute to the link between REM sleep and cardiometabolic traits. It is important to note that REM sleep in this study was inferred from accelerometer data, where classification relies in part on reduced movement (i.e., REM atonia). Serotonergic modulation, such as that induced by selective serotonin reuptake inhibitors (SSRIs), is known to increase motor tone during REM sleep, which may interfere with actigraphy-based REM detection without altering REM duration as measured by polysomnography. Thus, some genetic associations involving serotonin-related genes may reflect altered motor expression during REM rather than true differences in REM sleep architecture. Our findings are further supported by a recent study[19] identifying a regulatory variant at *HTR1F* associated with obstructive sleep apnea in non-obese individuals, where the same locus shows eQTL activity in adipose and heart tissues as well as in neurons. This work also links *HTR1F* variation to increased night-time arousals, reinforcing its role in sleep architecture and central sleep regulation. Additionally, we provide new insight into *MEIS1*'s phase-specific role in sleep architecture[9]. A likely causal variant (rs113851554, PIP = 100%) at 2p14, previously associated with chronotype, insomnia, and RLS[20], was linked to increased REM and decreased NREM sleep. *MEIS1* is highly constrained (pLI = 1 in gnomAD), consistent with its essential role in neuronal development and motor control. Although ascertainment of RLS in UK Biobank is imperfect, we performed additional analyses excluding both ICD-10 coded cases and probable RLS identified from the online questionnaire, removing 6,561 individuals ($\approx 8.2\%$ of the cohort). Associations at MEIS1 and other loci

remained robust across these exclusions and after removing short-sleepers ( < 7 h), supporting a role in REM/NREM regulation beyond movement disorders, while acknowledging that some residual under-diagnosis cannot be entirely ruled out. In addition, our pathway analyses highlighted enrichment for metal ion homoeostasis. This is noteworthy given the well-documented role of iron deficiency in restless legs syndrome, and the previous implication of MEIS1 and BTBD9 in iron metabolism. Iron supplementation is an established part of clinical management of RLS, supporting the biological plausibility of this pathway. Moreover, magnesium is often used by patients as an over-the-counter sleep aid, and although clinical evidence remains limited, experimental data suggest potential benefits for sleep quality. Together, these observations indicate that disturbances in metal ion balance may contribute to sleep regulation, and provide a possible mechanistic link between sleep architecture, RLS, and metabolic pathways.

Sex-stratified analyses identified loci associated with sleep traits in males and females, including *FOXP2*, *NRXN3*, and *LRP1B*. These genes are involved in neural development, hormonal signalling, or show sex-differential expression in brain tissue, supporting biological plausibility[21–23]. For example, FOXP2 is modulated by androgens and affects social behaviour in a sex-specific manner[21,22], while NRXN3 influences GABAergic inhibition differently in males and females[23]. Another notable example is TACR3, located near a female-specific intergenic signal for sleep efficiency, which encodes a receptor for neurokinin B and plays a critical role in hypothalamic regulation of puberty onset, neuroendocrine signalling, and cardiovascular control. Prior GWAS have rarely examined sex differences in sleep genetics; our findings support a sexually dimorphic architecture for REM and NREM sleep, meriting further investigation. It is important to note that these sex-specific findings were obtained in reduced sample sizes, and were not fine-mapped or functionally prioritised as in the main analyses. Thus, they should be viewed as exploratory, and confirmation in larger sex-stratified cohorts will be needed. In Mendelian randomisation analyses, genetically shorter night-time sleep was associated with higher BMI, HbA1c, and T2DM risk, supporting previous epidemiological and genetic studies[7,24,25]. Importantly, we also found suggestive causal effects of longer REM sleep on reduced risk of HF and CAD, outcomes for which the role of sleep architecture has been less well characterised[26,27]. Higher sleep efficiency was associated with lower BMI but higher WHRadjBMI, indicating divergent associations with total adiposity (BMI) versus central adiposity (WHRadjBMI).

Prior epidemiological studies using self-reported sleep duration have consistently described U-shaped associations with adverse health outcomes, with both short and long sleepers showing increased risk. Device-derived sleep duration differs from self-reported measures in important ways: it captures night-time sleep more precisely, reduces recall bias, and exhibits a narrower distribution with fewer extreme long-sleep values. As a result, device-based measures may attenuate or modify the classical U-shaped pattern, particularly at the upper end of the distribution. Our MR findings support detrimental effects of genetically shorter sleep on metabolic outcomes, consistent with the left-hand side of the U-shaped curve, but we did not observe evidence for adverse effects of genetically longer sleep. This discrepancy, also found in emerging observational studies of device-measured sleep duration[18], likely reflects differences between self-reported and accelerometer-derived traits, as well as the fact that lifelong genetic predisposition to longer sleep may not capture the same constructs as extended self-reported sleep associated with underlying morbidity or fatigue.

Our study benefits from the use of objectively measured sleep traits from wrist-worn accelerometers in a large sample (n = 80,013), and the exclusion of confounding factors such as shift work or daylight saving effects. Sleep stages were inferred using a deep neural network trained on over 1100 nights of polysomnography, ensuring robust

classification of REM and NREM periods. Here, we performed genome-wide association analyses focusing on REM and NREM sleep stages. Nonetheless, limitations include the predominantly European ancestry and relatively healthy, high-SES profile of UK Biobank participants[28]. Although this cohort is not fully representative, findings are generally reproducible in other populations[29]. Nevertheless, independent replication in larger and more diverse cohorts will be important to validate these associations. Moreover, REM/NREM estimates were indirectly inferred from movement data, potentially introducing measurement imprecision. One limitation of our study is that the accelerometer-based model used to estimate REM and NREM sleep exhibited systematic bias when compared to polysomnography: specifically, a mean underestimation of REM sleep by 17.1 min and an overestimation of NREM sleep by 31.1 min. The overall concordance with polysomnography was also modest (Cohen's kappa = 0.32), indicating only fair agreement. These factors may affect downstream associations, particularly for phase-specific traits, and should be considered when interpreting the results. However, the observed associations with age, sex, and BMI support the validity of the derived traits. We note the possibility of partial sample overlap between our UK Biobank analysis and the GWAS summary statistics used for some outcomes. While this does not affect LDSC estimates, it could introduce minor bias in MR analyses; however, the use of robust instruments and large independent outcome samples mitigates this concern.

In summary, we identified genetic loci associated with REM and NREM sleep and demonstrated phase-specific and sex-specific genetic influences on sleep architecture. These findings deepen our understanding of sleep genetics and its potential impact on metabolic and cardiovascular health. They also underscore the value of targeting both sleep duration and quality in preventive strategies aimed at reducing the burden of cardiometabolic disease. Importantly, the identification of candidate genes and the implication of regions with potential regulatory roles lay the groundwork for future functional studies to dissect causal mechanisms underlying sleep regulation.

## Methods

All research procedures were conducted in accordance with relevant ethical regulations. UK Biobank has obtained ethical approval from the UK Biobank Research Ethics Committee, and all participants provided informed consent. This study was conducted under UK Biobank application number 59070.

### Participants and sleep traits

The study population was drawn from the UK Biobank cohort, a large, longitudinal, population-based study involving over 500,000 individuals[30]. For the present analyses, we focused on a subset of 103,680 participants who wore a wrist-worn accelerometer for 7 days between 2013 and 2015[31] and restricted the sample to individuals of European ancestry. Participants were excluded if their data could not be calibrated, exhibited unrealistically high values (e.g., average vector magnitude > 100 mg), or had insufficient wear-time (i.e., less than 3 days or no recorded wear in every hour of the 24-hour period). Additional exclusions were applied to shift workers and individuals with data impacted by daylight saving time transitions (Supplementary Fig. 1).

Sleep traits, including night-time sleep duration, sleep efficiency, and the durations of REM and NREM sleep, were derived using a self-supervised deep recurrent neural network for sleep stage classification, trained on over 1,100 nights of concurrent laboratory-based polysomnography and accelerometry data. The Kappa score for REM/NREM/wake classification was 0.32. The model's predictions and polysomnography data show a difference of 48.2 min for sleep duration, −17.1 min for REM duration, 31.1 min for NREM duration, and 9.2% for sleep efficiency[18]. While the model showed improved performance over previous approaches using accelerometry, concordance with

polysomnography was modest. As such, the REM and NREM traits derived here reflect estimates based on movement patterns, and may not fully correspond to PSG-defined sleep stages.

### Genotyping, imputation, and quality control

Genotyping and imputation procedures for UK Biobank participants were performed centrally by UK Biobank using standard protocols, as described in Bycroft et al.[30].

In brief, genotyping was performed using the Affymetrix UK BiLEVE Axiom array for the initial ~50,000 participants and the Affymetrix UK Biobank Axiom Array for approximately 450,000 participants. The two arrays are over 95% similar and together cover approximately 820,000 SNP and indel markers (http://www.ukbiobank.ac.uk/). Quality control and imputation of over 90 million SNPs, indels, and large structural variants were carried out centrally. Following imputation, additional SNP-level quality control was performed locally, excluding variants with an imputation INFO score <0.3 or a minor allele frequency (MAF) < 0.01. Individual-level quality control was also conducted to remove samples identified as outliers based on heterozygosity, missingness, mismatched sex (where discrepancies between reported and genetically inferred sex were detected, indicating potential sample processing errors), or sex chromosome aneuploidy (Supplementary Fig. 1).

### Genome-wide association analyses

All association tests were performed using REGENIE v.3.1.2[32], which implements a linear mixed model (LMM) to account for population structure and relatedness, allowing for the inclusion of related individuals and thus enhancing the power to detect genetic associations. We analysed autosomal and X chromosome data assuming an additive genetic model, adjusted for age at accelerometry, sex, study centre, season of activity monitor wear, and genotyping array. Additionally, the first 10 principal components (PCs) of the white European subset included in the analysis were incorporated as covariates to control for subtle differences in ancestry. For the X chromosome analysis, male genotypes were coded as diploid (0,2).

To further improve statistical power in detecting genetic loci associated with sleep, we performed multitrait analysis of genome-wide association studies (MTAG) alongside single-trait GWAS. MTAG performs a meta-analysis of GWAS summary statistics across genetically correlated traits, addressing sample overlap to enhance power and precision[33].

### Sensitivity analyses

To evaluate whether observed associations for individual variants were influenced by specific subgroup characteristics, we performed several sensitivity analyses: (1) stratification by sex (males only, females only) (2) adjustment for body mass index (BMI; UK Biobank data field 21001), (3) exclusion of individuals with ICD-10 coded movement disorders (Supplementary Data 14), (4) exclusion of individuals taking medications known to influence sleep (Supplementary Data 14); (5) exclusion of participants with night-time sleep duration <7 h; and (6) exclusion of individuals with probable restless legs syndrome identified via the UK Biobank online sleep questionnaire in combination with ICD-10 coded cases (Supplementary Data 14).

### Genomic loci characterisation, linkage disequilibrium score regression, and fine-mapping

Independent genome-wide significant signals were identified using an LD-based clumping procedure implemented in PLINK v.1.9[34]. This procedure clusters SNPs based on their linkage disequilibrium (LD) with nearby SNPs and their P values. We used the following parameters: index SNP P value threshold of $5 \times 10^{-8}$ (--clump-p1) and clumped SNP P value threshold of 0.05 (--clump-p2), with a physical distance threshold of 1000 kb (--clump-kb) and an LD threshold of 0.01

(--clump-r2). SNPs within each clump were represented by the index SNP. Variant annotation was performed using the Ensembl Variant Effect Predictor (VEP) tool[35].

To identify additional independent signals in regions of association, conditional SNP association analysis was conducted using the Genome-wide Complex Trait Analysis (GCTA) v.1.24.4 tool[36]. Any lead SNPs within known high-LD regions on chromosomes 6 (28.48–33.45 MB) and 17 (43.5–45.5 MB) were treated as a single locus in the GCTA analysis.

Independent genome-wide significant signals were classified as novel if the lead SNP was not in LD ($r^2 \geq 0.1$) with any previously reported variant associated with the same sleep trait. Previously reported associations were identified through the GWAS Catalog and cross-referenced with Open Targets Genetics[37], which integrates results from UK Biobank, FinnGen, and other sources.

LD score (LDSC) regression was performed for each GWAS to estimate SNP-based heritability, evaluate test statistic inflation, and assess genetic correlations between traits. The regression was performed by regressing the GWAS test statistics ($\chi^2$) onto each SNP's LD score, which is the sum of squared correlations between the MAF of the SNP and those of all other SNPs. This regression allows for the estimation of heritability from the slope and can detect residual confounders through the intercept. The LDSC regression intercept accounts for inflation in the $\chi^2$ statistics that is not due to stratification or other confounding factors. Inflation caused by polygenicity will correlate with LD, while inflation from stratification or relatedness will not[38]. LD scores and weights were downloaded from LDSC repository (https://data.broadinstitute.org/alkesgroup/LDSCORE) for European populations.

GWAS summary statistics were standardised using the munge function implemented in LDSC. To minimise bias from poorly imputed SNPs, an imputation quality score threshold of > 0.9 was applied. SNPs whose alleles did not match those in the 1000 Genomes reference or were strand-ambiguous were excluded.

Fine-mapping was performed using the "Sum of Single Effects" (SuSiE) Bayesian model[39], which is a stepwise conditional analysis that improves upon traditional methods by accounting for the uncertainty in selecting associated SNPs. This approach allows for the detection of multiple causal signals and identifies causal variants even when the SNP with the lowest P value is not the causal variant. To reduce the complexity of the analysis and focus on the most relevant variants, we limited the SNPs to those within a 500 kb window surrounding the lead variant at each locus. In-sample dosage-based LD matrices were computed using PLINK v.1.9[34] and 95% credible sets (CS) were evaluated. A SNP was considered as a likely causal variant if the posterior inclusion probability (PIP) was ≥ 0.80, indicating a high probability of being associated with the trait. This threshold was chosen to prioritise variants with strong evidence of association, although the method does not provide definitive proof of causality.

### Functional downstream analyses
To conduct in silico downstream functional analyses of GWAS results, we used the FUMA (Functional Mapping and Annotation) GWAS platform v.1.5.2[40] which implements the MAGMA (Multi-Marker Analysis of GenoMic Annotation) framework v.1.08[41].

First, functional annotation of all genome-wide significant SNPs and SNPs in LD with them ($r2 \geq 0.6$) was performed using Annotate Variation (ANNOVAR) enrichment test (gene-based annotation), which annotates the functional consequence of SNPs on Ensemble (v102) protein coding genes (e.g. intron and exon). Functionally annotated SNPs were subsequently mapped to genes using three strategies: positional mapping (physical distance), expression quantitative trait loci (eQTL) mapping (eQTL association), and chromatin interaction.

For positional mapping, based on annotations obtained from ANNOVAR, the physical distance of each SNP from known protein-coding genes was set to the default window of 10 kb in the human reference assembly (hg19/GRCH37).

eQTL mapping was used to map independent significant SNP and SNPs in LD with them to genes that show a significant association with cis-eQTLs (i.e., allelic variation of the SNP associated with expression levels of the gene). This approach maps SNPs to genes located up to 1MB apart. For the eQTL mapping, we used significant cis-eQTLs from GTEx v8 across the following tissues: Adipose (Subcutaneous; Visceral/Omentum), Adrenal Gland, Artery (Aorta; Coronary; Tibial), Blood (Cells-EBV-transformed lymphocytes; Whole blood), Brain (Amygdala; Anterior cingulate cortex (BA24); Caudate (basal ganglia); Cerebellar Hemisphere; Cerebellum; Cortex; Frontal Cortex (BA9); Hippocampus; Hypothalamus; Nucleus accumbens (basal ganglia); Putamen (basal ganglia); Spinal cord (cervical c-1); Substantia nigra), Breast-Mammary Tissue, Colon (Sigmoid; Transverse), Oesophagus (Gastroesophageal Junction; Mucosa; Muscularis), Heart (Atrial Appendage; Left Ventricle), Kidney-Cortex, Liver, Lung, Muscle-Skeletal, Nerve-Tibial, Ovary, Pancreas, Pituitary, Prostate, Minor Salivary Gland, Cells-Cultured fibroblasts, Skin (Not Sun Exposed-Suprapubic; Sun Exposed-Lower leg), Small Intestine-Terminal Ileum, Spleen, Stomach, Testis, Thyroid, Uterus, Vagina. Significance followed the GTEx v8 panel definition of significant cis-eQTLs (FDR q < 0.05), as implemented in FUMA. In addition to GTEx v8, to strengthen the biological interpretation of sleep-associated loci in brain tissue, we interrogated brain-specific eQTL datasets including PsychENCODE, CMC (DLPFC, with and without SVA correction), BrainSeq, BRAINEAC, and xQTLServer.

Chromatin interaction mapping was performed by overlapping independent significant SNPs and SNPs in LD with them with one end of significantly interacting regions in tissue/cell types. These SNPs were then mapped to genes whose promoter regions (by default, 250 bp upstream and 500 bp downstream of the transcription start site) overlap with the other end of the significant interactions. This mapping can involve long-range interactions as there is no distance boundary. FUMA uses data on the 3D structure of chromatin interactions from Hi-C data for 23 tissues and cell types, as well as tissue and cell type data from FANTOM and PsychENCODE. The significance threshold was defined as $FDR < 1.0 \times 10^{-6}$ in FUMA, based on prior recommendations[42].

Furthermore, MAGMA was used to perform gene-based, gene-set, and gene-property (tissue gene expression) analyses of the full GWAS summary results. In brief, gene-based analysis computes gene-based P values for SNPs mapped to protein-coding genes, using a SNP-wise mode that aggregates SNP P values into a gene-level test statistic. Gene boundaries were determined using NCBI build 37, encompassing 18,877 protein-coding genes. SNPs located within gene boundaries were included in the analysis to derive P values reflecting associations with sleep traits. Linkage disequilibrium within and between genes was assessed using the UK Biobank release2b 10 K White British reference panel. A Bonferroni correction was applied to account for multiple testing, setting the genome-wide significance threshold at $P \leq 6.6 \times 10^{-7}$, considering 18,877 genes and four sleep traits.

A competitive gene-set analysis was performed to investigate whether the polygenic signal from the GWASs clustered in specific biological pathways. Competitive tests control for Type 1 error rate and assess whether genes within the set are more strongly associated with sleep traits compared to other genes (39). A total of 10,894 gene-sets from Gene Ontology (40), Reactome (41), and SigDB (42) were analysed for enrichment in sleep measures. Bonferroni correction was again applied to account for the multiple tests across the 10,894 gene sets.

Gene-property analysis was conducted to explore whether tissue-specific expression levels in 30 broad tissue types and 53 specific tissues were predictive of a gene's association with sleep traits. Tissue types were sourced from the GTEx v8 RNA-seq database (43), and

expression values were log2 transformed with a pseudocount of 1 after winsorising at 50, with the average expression value taken from each tissue. Multiple testing was controlled for using Bonferroni correction. In addition, we performed exploratory cell-type enrichment analyses using the extensive catalogue of single-cell reference datasets available in FUMA (including adult and developmental human brain as well as selected peripheral tissues). Enrichment was assessed at the cell-type level, and p-values were adjusted for multiple comparisons using Bonferroni correction, as implemented in FUMA.

## eQTL colocalization analysis

To assess whether the same variant influences both sleep traits and gene expression, a Bayesian colocalization analysis was performed using GTEx v8 eQTL data. Analyses were conducted with the Coloc method[43], implemented in the xQTLbiolinks R package[44] (https://github.com/lilab-bioinfo/xQTLbiolinks). Sentinel SNPs were identified for each GWAS using the xQTLanalyze_getSentinelSnp function with parameters $P < 5.0 \times 10^{-8}$ and SNP-to-SNP distance > 1 Mb. Genes within 1 Mb of sentinel SNPs were considered for analysis (Supplementary Data 15). The GWAS genome version was converted from GRCh37 to GRCh38 for compatibility with GTEx v8.

We assumed a prior probability that a SNP is associated with either sleep or gene expression (default = $1.0 \times 10^{-4}$), and both GWAS and gene expression (default = $1.0 \times 10^{-5}$) for all Coloc analyses. These priors were initially recommended for the analysis of eQTL data[45] and are widely adopted in applied practice. The Coloc method calculates posterior probabilities for five hypotheses, including the hypothesis of shared causal variants between sleep traits and gene expression. A posterior probability (PP.H4 ≥ 75%) was used as a threshold to indicate strong evidence for colocalization, leading to the prioritisation of genes within the corresponding loci. A comprehensive list of tissues, including brain regions, was investigated (Supplementary Data 16).

## LDSC genetic correlation, Mendelian randomisation, and pairwise colocalization analysis

To explore the genetic overlap between sleep traits and ten related diseases/traits – type 2 diabetes mellitus (T2DM), haemoglobin A1c (HbA1c), coronary artery disease (CAD), heart failure (HF), body mass index (BMI), waist-to-hip ratio adjusted for BMI (WHRadjBMI), biological age (an estimate of an individual's age based on biological markers, reflecting overall health status and aging process), memory performance, Alzheimer's disease (AD), and Parkinson's disease (PD) – we estimated genetic correlations using LDSC. These traits were chosen for their known or hypothesised links with sleep, particularly concerning cardiometabolic health, aging, and neurodegenerative diseases.

The purpose was to identify potential biological connections that could be explored further with causal inference methods like Mendelian randomisation (MR).

To further explore the potential relationships, including possible causal links, between sleep traits and these ten diseases/traits, we conducted a two-sample bidirectional MR analysis. Instrumental variables (IVs) were identified from our sleep GWAS. We included only independent SNPs ($r^2 < 0.01$ and window size = 1000 kb) that were genome-wide significant ($P < 5 \times 10^{-8}$) and had a minor allele frequency > 0.01. SNPs with weak statistical power (F statistics < 10) were excluded[46].

For the outcome datasets, we used publicly available summary-level GWAS data for sleep-related diseases and traits in populations of European ancestry. Where exposure-associated SNPs were unavailable in the outcome dataset, correlated SNPs ($r^2 \geq 0.9$) were used as substitutes. Effect alleles were harmonised across datasets, and incompatible or palindromic SNPs were excluded. Data sources and the list of IVs are summarised in Supplementary Data 17.

The primary MR analysis was performed using the inverse variance weighted (IVW) method with a random-effects model to combine Wald ratios for each SNP[47]. MR–Egger[48] and weighted median[49] methods were used as complementary approaches to assess directional pleiotropy. The MR-Egger intercept was used to estimate directional pleiotropy, with the slope providing a valid estimate when pleiotropy was detected. The weighted median method was applied as it provides valid causal estimates even when up to 50% of the instruments are invalid. The MR-PRESSO[50] global test and MR robust adjusted profile score (MR.RAPS)[51] methods were applied to account for pleiotropy and to correct for heterogeneity in instrument-exposure associations and outlier SNPs, thus reducing bias from these sources. MR Steiger filtering was used to assess the most likely direction of effect for each SNP. Outliers and SNPs that were inconsistent with the expected direction of effect were removed, and MR causal estimates were re-evaluated. If only one SNP or fewer than three SNPs were available, the Wald ratio and fixed effects IVW were utilised, respectively. Cochrane's Q-statistic was used to assess the heterogeneity of SNP effects, with smaller P values indicating higher heterogeneity and potential for directional pleiotropy. This test was consistently applied across all analyses, with caution exercised in cases where the number of SNPs was limited, to avoid over-interpreting results with low statistical power. Bonferroni correction was applied to adjust for multiple comparisons, with a P value threshold of <0.005 for the ten diseases/traits used as exposures and <0.012 for the four sleep traits used as exposures, to support evidence for a potential association between sleep traits and each outcome. To be considered robust, results were required to show a statistically significant association ($P < 0.005$ or 0.012, as per Bonferroni correction) using IVW, with directionally concordant evidence from secondary methods. Results were further deemed reliable if the MR–Egger intercept and MR-PRESSO global test were non-significant ($P > 0.05$), indicating no evidence of directional pleiotropy or outlier bias. Discrepancies in directionality observed with MR–Egger were interpreted cautiously, as they may reflect the method's lower statistical power rather than pleiotropic bias. P values between 0.005 and 0.05 or between 0.012 and 0.05 were regarded as suggestive evidence of an association.

All MR analyses were performed using R (version 4.2.0) and the "TwoSampleMR" package (version 0.5.6), with MR-PRESSO and RAPS analyses conducted using the "MRPRESSO" and "MR.raps" R packages.

Approximate Bayes factor (ABF) colocalization analysis was used to assess shared genetic causal variants between sleep traits and related health outcomes. Colocalization was performed using the Coloc method, which allows for the detection of shared causal variants between traits at the same locus[52]. A posterior probability of PP.H4 ≥ 75% was considered suggestive of colocalization, indicating a common genetic mechanism driving both traits. Sensitivity analyses were performed to examine the impact of prior probabilities on our results.

## Reporting summary

Further information on research design is available in the Nature Portfolio Reporting Summary linked to this article.

# Data availability

This research was conducted using the UK Biobank resource (application number 59070). Individual-level UK Biobank data analysed in this study are subject to access restrictions in accordance with UK Biobank data governance policies and participant consent and cannot be made publicly available. Individual-level data are available to bona fide researchers upon application to UK Biobank (https://www.ukbiobank.ac.uk/enable-your-research/apply-for-access), subject to approval by the UK Biobank Access Management Team and execution of a data access agreement.

The GWAS summary statistics generated in this study have been deposited in the GWAS Catalog under accession numbers

GCST90824104, GCST90824105, GCST90824106, and GCST90824107 (GCP001581). These summary statistics will be made publicly available upon publication and will be accessible via the GWAS Catalog (https://www.ebi.ac.uk/gwas/).

Brain-specific eQTL datasets including PsychENCODE, CMC, BrainSeq and BRAINEAC were accessed via the FUMA GWAS platform (https://fuma.ctglab.nl/), which integrates publicly available functional annotation resources. GTEx v8 data were obtained from the GTEx Portal (https://gtexportal.org/). LD scores and weights were downloaded from the LDSC repository (https://data.broadinstitute.org/alkesgroup/LDSCORE/). These resources are publicly available according to the terms specified by their respective repositories. The sources of GWAS summary statistics used for Mendelian randomisation analyses, including PMIDs and references to the original studies, are described in Supplementary Data 17. All other data supporting the findings of this study are available within the Article and its Supplementary Information files.

## Code availability
No novel methodological code was developed for this study. All analyses were conducted using publicly available software tools and relevant parameters are described in the Methods section. Analysis scripts used to run standard software pipelines are available from the corresponding author upon reasonable request.

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

## Acknowledgements

This research was supported by Novo Nordisk. A.D. acknowledges funding from the Wellcome Trust (223100/Z/21/Z), Novo Nordisk, Swiss Re, Health Data Research UK, and the British Heart Foundation Centre of Research Excellence (RE/18/3/34214). A.D. also acknowledges support for attendance at scientific meetings and a research donation from Swiss Re supporting accelerometer data collection in the China Kadoorie Biobank. The authors thank the participants of the UK Biobank and the UK Biobank research teams for data collection at baseline and during the accelerometry study. The authors also thank Maylor B.D. and Chan S. for their support with additional analyses conducted during the revision process.

## Author contributions

L.P. performed the analyses, interpreted the results, and wrote the manuscript. H.Y. developed and provided the sleep phenotypes. A.D. and J.M.M.H. conceived the study and supervised the project. S.D.K. and D.R. contributed to the interpretation of the results. L.C., K.S.-B., and S.v.D. provided scientific input and contributed to manuscript revision. All authors reviewed and approved the final version of the manuscript.

## Competing interests

A.D. is supported by grants from the Wellcome Trust [223100/Z/21/Z], Novo Nordisk, Swiss Re, Health Data Research UK, and the British Heart Foundation Centre of Research Excellence [RE/18/3/34214]; has accepted consulting fees from the University of Wisconsin (NIH R01 grant) and Harvard University (NIH R01 grant); received support for presentations or attendance at several conferences; and has received a donation from SwissRe for accelerometer data collection in the China Kadoorie Biobank. LP is supported by Novo Nordisk. LC and JMMH are full-time employees of Novo Nordisk and hold shares in Novo Nordisk A/S. The other authors declare no competing interests.
