## [Transparent Peer Review File · Nature Communications]

Genetic architecture of sleep in a genome wide association study of device measured sleep traits

Corresponding Author: Dr Laura Portas

Version 0:

Reviewer comments:

Reviewer #1

(Remarks to the Author)

The authors have carefully and clearly addressed all suggestions and comments that I provided in the review. I have no additional comments for this manuscript. There is sufficient detail in each point addressed. Additionally, the authors have provided additional analyses to address the questions.

Reviewer #2

(Remarks to the Author)

Overall, the authors were responsive to reviewers, and responded well to the two primary concerns raised by all three reviewers (accuracy of the deep learning model and potential confound of actigraphy with RLS and other factors).

One minor comment - I did misinterpret the Alzheimer connection. To prevent others from making the same mistake, perhaps revise

"REM sleep showed negative genetic correlations with several cardiometabolic traits 211 (HF, CAD, T2DM, BMI) and biological age, but a positive correlation with Alzh]

To something like

To "REM sleep showed negative correlations with estimated genetic risk of several cardiometabolic traits 211 (HF, CAD, T2DM, BMI) and biological age, but a positive correlation with genetic risk of Alzh. . .

Reviewer #3

(Remarks to the Author)

Portas et al. reported the first GWAS of device-derived REM and NREM sleep stages, as well as sleep duration and sleep efficiency. This work has the potential to advance our understanding of the molecular background of human sleep architecture and its links to complex diseases. However, several major concerns remain that limit the current impact of the study:

1. Consistent with other reviewers, I have substantial concerns regarding the accuracy of the sleep stage classification. The model described shows notable bias, which represents a key limitation and may compromise the validity of the genetic findings. Replication using polysomnography-derived sleep stages would be highly valuable. Although the authors state that sufficiently large PSG datasets are unavailable, replication in small samples with PSG data is still desired.

2. This study has some overlaps with previous UK Biobank accelerometry-based GWAS (Doherty et al., 2018; Jones et al., 2019). The authors state that a different algorithm was used to derive sleep duration and efficiency. It would be helpful to clearly describe how this algorithm differs from prior approaches (e.g., using GGIR), and to report comparative performance

and correlations between the metrics using different methods.

3. Additional details on phenotype harmonization are needed. Beyond the QC steps described, were missing values imputed? What is the distribution of the derived traits? Some normalization or winsorization may be necessary to address skewness and outliers.

4. Prior studies using self-reported sleep duration have demonstrated U-shaped associations with complex disease risk. It is unclear whether device-derived sleep duration and/or stage-specific duration shows similar patterns. While formally testing this may be beyond the scope of the current manuscript, it would be helpful for the authors to discuss the relevance of these known associations in the context of their phenotypes and findings.

Version 1:

Reviewer comments:

Reviewer #3

(Remarks to the Author)

The authors have sufficiently addressed my concerns, and I have no additional comments on the study. Please ensure that the URL for the summary statistics is provided in the final published paper.

REVIEWER COMMENTS

Reviewer #1 (Remarks to the Author):

The authors have carefully and clearly addressed all suggestions and comments that I provided in the review. I have no additional comments for this manuscript. There is sufficient detail in each point addressed. Additionally, the authors have provided additional analyses to address the questions.

We thank the reviewer for the positive feedback and are pleased to hear that the revisions and additional analyses adequately addressed all previous comments. We appreciate the reviewer's time and constructive input throughout the review process.

Reviewer #2 (Remarks to the Author):

Overall, the authors were responsive to reviewers, and responded well to the two primary concerns raised by all three reviewers (accuracy of the deep learning model and potential confound of actigraphy with RLS and other factors).

One minor comment - I did misinterpret the Alzheimer connection. To prevent others from making the same mistake, perhaps revise "REM sleep showed negative genetic correlations with several cardiometabolic traits (HF, CAD, T2DM, BMI) and biological age, but a positive correlation with Alzh.." to something like "REM sleep showed negative correlations with estimated genetic risk of several cardiometabolic traits (HF, CAD, T2DM, BMI) and biological age, but a positive correlation with genetic risk of Alzhem..."

We thank the reviewer for this helpful suggestion. We agree that clarifying the phrasing will prevent potential misinterpretation. We have revised the text accordingly to:

"REM sleep showed negative genetic correlations with estimated genetic risk of several cardiometabolic traits (HF, CAD, T2DM, BMI) and with biological age, but a positive correlation with genetic risk of Alzheimer's disease."

Reviewer #3 (Remarks to the Author):

Portas et al. reported the first GWAS of device-derived REM and NREM sleep stages, as well as sleep duration and sleep efficiency. This work has the potential to advance our understanding of the molecular background of human sleep architecture and its links to complex diseases. However, several major concerns remain that limit the current impact of the study:

1. Consistent with other reviewers, I have substantial concerns regarding the accuracy of the sleep stage classification. The model described shows notable bias, which represents a key limitation and may compromise the validity of the genetic findings. Replication using polysomnography-derived sleep stages would be highly valuable. Although the authors state that sufficiently large PSG datasets are unavailable, replication in small samples with PSG data is still desired.

We fully agree with the reviewer that PSG represents the current best standard for sleep stage classification. We also recognise that the accelerometer-based model has systematic bias relative to PSG, which we explicitly acknowledge and discuss in the manuscript.

However, to our knowledge – and following direct consultation with several leading investigators in the field – there are currently no PSG datasets that (i) include REM and NREM sleep duration derived in a standardised manner, (ii) are genotyped, and (iii) are sufficiently powered to replicate genome-wide association signals.

Existing PSG cohorts typically include only a few hundred participants, are often clinically ascertained (e.g., sleep apnea or narcolepsy), and do not include the same REM/NREM phenotypes that were analysed in our study. Furthermore, most published PSG-based genetic studies focus on candidate genes or specific sleep disorders rather than performing genome-wide analyses of sleep architecture.

For these reasons, a formal replication using PSG-derived traits is not feasible at present.

Nevertheless, we have taken several steps to address the concern around sleep stage classification accuracy:

- We explicitly discuss this limitation in the manuscript (Discussion; Methods), including detailed quantification of model bias and concordance with PSG.
- Sensitivity analyses show that the core genetic signals remain robust after excluding individuals with movement disorders, those taking sleep-influencing medications, and short sleepers, reducing the likelihood that model bias in relation to these traits drives the associations.
- The identified loci are biologically coherent, mapping to genes with established roles in sleep, circadian regulation, movement disorders, or metabolic pathways.

We remain committed to pursuing PSG-based replication as soon as suitable datasets become available.

2. This study has some overlaps with previous UK Biobank accelerometry-based GWAS (Doherty et al., 2018; Jones et al., 2019). The authors state that a different algorithm was used to derive sleep duration and efficiency. It would be helpful to clearly describe how this algorithm differs from prior approaches (e.g., using GGIR), and to report comparative performance and correlations between the metrics using different methods.

We thank the reviewer for this helpful suggestion and agree that a direct comparison with prior accelerometry-based approaches is important. To address this point, we re-derived sleep duration and sleep efficiency in the same UK Biobank subset using the standard GGIR pipeline, which underpins previous large-scale analyses, and compared these estimates with those obtained from our deep-learning-based model.

For sleep duration, we observed a strong correlation between methods (Pearson $r = 0.71$), indicating good overall agreement despite methodological differences. GGIR-derived sleep duration estimates were on average shorter than those from our model (mean difference -20.3 minutes, SD 44.3 minutes). This is consistent with the fact that both approaches use similar principles to identify the main sleep window, but our method includes a subsequent refinement stage that more precisely identifies sleep and wake segments within this main sleep window.

In contrast, sleep efficiency showed weaker agreement between methods ($r = 0.30$), with GGIR yielding systematically higher values (mean difference $+0.069$, SD 0.085). This is not unexpected given that sleep efficiency is a more challenging phenotype to infer from wrist accelerometry and that the two approaches differ substantially in their modelling strategies. GGIR relies on a random forest-based heuristic framework, whereas our approach uses a recurrent neural network trained on polysomnography-labeled data. Importantly, our model has been validated on a substantially larger number of individuals ($n \approx 1,300$ versus ≈ 150 in earlier work) and shows acceptable within-person reliability for sleep efficiency ($r = 0.58$), as reported previously.

For completeness, we attach Bland-Altman plots and scatterplots comparing GGIR- and deep-learning-derived sleep duration and sleep efficiency in this response.

Bland-Altman and scatterplot for sleep duration

Bland-Altman and scatterplot for sleep efficiency

3. *Additional details on phenotype harmonization are needed. Beyond the QC steps described, were missing values imputed? What is the distribution of the derived traits? Some normalization or winsorization may be necessary to address skewness and outliers.*

No imputation was applied to the sleep traits. REM, NREM, sleep duration and sleep efficiency were derived only from accelerometer windows passing predefined quality checks, and participants with insufficient wear-time, shift work, daylight-saving transition, or other measurement issues were excluded.

After these exclusions, all four traits showed physiologically plausible ranges and distributions, consistent with expectations for a large population-based cohort of older adults and highly concordant with prior accelerometer-based studies in the UK Biobank. The distributions were symmetric or only moderately skewed in the expected direction (sleep duration: -0.50 ; sleep efficiency: -1.06 ; REM: 0.40 ; NREM: 0.07), values that are typical of continuous sleep phenotypes in population cohorts. Minimum and maximum values fell within realistic physiological limits (e.g., sleep duration 60-763 min; REM 0.17-269 min; NREM 43-660 min; sleep efficiency 24-98%), and the observed sex differences were biologically expected.

Inspection of the 0.1-99.9% percentiles confirmed that all traits lay within plausible physiological ranges.

As an additional robustness check, we had previously conducted analyses that excluded participants with <7 hours of average night-time sleep, to minimise the influence of atypically short-sleep patterns; we found that lead loci and effect directions remained unchanged, further supporting the stability of the phenotypes.

Given the robustness of REGENIE's mixed-model framework to moderate deviations from normality, we did not apply transformation or winsorization in the GWAS analyses.

4. Prior studies using self-reported sleep duration have demonstrated U-shaped associations with complex disease risk. It is unclear whether device-derived sleep duration and/or stage-specific duration shows similar patterns. While formally testing this may be beyond the scope of the current manuscript, it would be helpful for the authors to discuss the relevance of these known associations in the context of their phenotypes and findings.

We thank the reviewer for this insightful comment. Prior epidemiological studies based on self-reported sleep duration have indeed reported U-shaped associations with several health outcomes. We agree that contextualising our findings in light of this literature is important. We have now added a dedicated paragraph to the Discussion (page 16) describing how device-derived sleep duration differs from self-reported measures, how these differences may attenuate or modify the classical U-shaped pattern, and how our MR findings relate to the established evidence for short- and long-sleep associations. This addition clarifies the relevance of prior U-shaped epidemiological associations in the context of our phenotypes and results.

Text added to the discussion section: *"Prior epidemiological studies using self-reported sleep duration have consistently described U-shaped associations with adverse health outcomes, with both short and long sleepers showing increased risk. Device-derived sleep duration differs from self-reported measures in important ways: it captures night-time sleep more precisely, reduces recall bias, and exhibits a narrower distribution with fewer extreme long-sleep values. As a result, device-based measures may attenuate or modify the classical U-shaped pattern, particularly at the upper end of the distribution. Our MR findings support detrimental effects of genetically shorter sleep on metabolic outcomes, consistent with the left-hand side of the U-shaped curve, but we did not observe evidence for adverse effects of genetically longer sleep. This discrepancy, also found in emerging observational studies of device-measured sleep duration¹⁸, likely reflects differences between self-reported and accelerometer-derived traits, as well as the fact that lifelong genetic predisposition to longer sleep may not capture the same constructs as extended self-reported sleep associated with underlying morbidity or fatigue."*

REVIEWER COMMENTS

Reviewer #3 (Remarks to the Author):

The authors have sufficiently addressed my concerns, and I have no additional comments on the study. Please ensure that the URL for the summary statistics is provided in the final published paper.

We have updated the Data Availability statement in the manuscript to include the GWAS Catalog accession numbers (GCST90824104-GCST90824107) and the corresponding URL to the GWAS Catalog (<https://www.ebi.ac.uk/gwas/>). The summary statistics will be publicly accessible via the GWAS Catalog upon publication.